# Nuclear response to divergent mitochondrial DNA genotypes modulates the interferon immune response

M. Isabel G. Lopez Sanchez[1,2]*, Mark Ziemann[3,4], Annabell Bachem[5], Rahul Makam[1], Jonathan G. Crowston[1,2], Carl A. Pinkert[6], Matthew McKenzie[4,7,8], Sammy Bedoui[5], Ian A. Trounce[1,2]*

1 Centre for Eye Research Australia, Royal Victorian Eye and Ear Hospital, Melbourne, Victoria, Australia, 2 Ophthalmology, Department of Surgery, University of Melbourne, Melbourne, Victoria, Australia, 3 Department of Diabetes, Monash University Central Clinical School, The Alfred Medical Research and Education Precinct, Melbourne, Victoria, Australia, 4 School of Life and Environmental Sciences, Deakin University, Victoria, Australia, 5 Department of Microbiology and Immunology, The University of Melbourne at the Peter Doherty Institute for Infection and Immunity, Melbourne, Victoria, Australia, 6 Department of Pathobiology, College of Veterinary Medicine, Auburn University, Auburn, Alabama, United States of America, 7 Centre for Innate Immunity and Infectious Diseases, Hudson Institute of Medical Research, Melbourne, Victoria, Australia, 8 Department of Molecular and Translational Science, Monash University, Melbourne, Victoria, Australia

* isabel.lopez@unimelb.edu.au (MIGLS); i.trounce@unimelb.edu.au (IAT)

**Data Availability Statement:** The datasets generated and/or analyzed during the current study are available in the GEO repository, under

## Abstract

Mitochondrial OXPHOS generates most of the energy required for cellular function. OXPHOS biogenesis requires the coordinated expression of the nuclear and mitochondrial genomes. This represents a unique challenge that highlights the importance of nuclear-mitochondrial genetic communication to cellular function. Here we investigated the transcriptomic and functional consequences of nuclear-mitochondrial genetic divergence *in vitro* and *in vivo*. We utilized xenomitochondrial cybrid cell lines containing nuclear DNA from the common laboratory mouse *Mus musculus domesticus* and mitochondrial DNA (mtDNA) from *Mus musculus domesticus*, or exogenous mtDNA from progressively divergent mouse species *Mus spretus*, *Mus terricolor*, *Mus caroli* and *Mus pahari*. These cybrids model a wide range of nuclear-mitochondrial genetic divergence that cannot be achieved with other research models. Furthermore, we used a xenomitochondrial mouse model generated in our laboratory that harbors wild-type, C57BL/6J *Mus musculus domesticus* nuclear DNA and homoplasmic mtDNA from *Mus terricolor*. RNA sequencing analysis of xenomitochondrial cybrids revealed an activation of interferon signaling pathways even in the absence of OXPHOS dysfunction or immune challenge. In contrast, xenomitochondrial mice displayed lower baseline interferon gene expression and an impairment in the interferon-dependent innate immune response upon immune challenge with herpes simplex virus, which resulted in decreased viral control. Our work demonstrates that nuclear-mitochondrial genetic divergence caused by the introduction of exogenous mtDNA can modulate the interferon immune response both *in vitro* and *in vivo*, even when OXPHOS function is not compromised. This work may lead to future insights into the role of mitochondrial genetic

accession number GSE113340. Descriptive statistics for the experiments conducted in our study are included in Supporting Information File 1.

**Funding:** This research was funded by the National Health and Medical Research Council APP115979 (Dr Ian A. Trounce), the Mito Foundation (Dr M. Isabel G. Lopez Sanchez), and the State Government of Victoria, Operational Infrastructure Support (Centre for Eye Research Australia). The funders had no role in study design, data collection and analysis, decision to publish, or preparation of the manuscript.

**Competing interests:** The authors have declared that no competing interests exist.

variation and the immune function in humans, as patients affected by mitochondrial disease are known to be more susceptible to immune challenges.

## Introduction

Mitochondria generate most of the energy required for cellular function via the oxidative phosphorylation (OXPHOS) system. Biogenesis of OXPHOS protein complexes requires the coordinated expression and assembly of protein subunits encoded in the nuclear and mitochondrial genomes [1]. This represents a unique challenge, whereby genetic changes in either genome can impact the energy-generating capacity of mitochondria [2] and other mitochondrial functions.

Despite growing evidence of the role of nuclear-mitochondrial genetic interactions in metabolism and their contribution to discrete phenotypic changes [3–6], research models are limited due to the complexity associated with manipulating the mitochondrial genome directly [7]. Nuclear-mitochondrial genetic interactions have been studied in conplastic mice by introducing donor mtDNA into recipient strains with successive backcrossing for ten or more generations [3]. Alternatively, we and others have generated xenomitochondrial (also referred to as transmitochondrial) cytoplasmic hybrids (cybrids), whereby cells chemically-devoid of mtDNA are fused with enucleated cells for homoplasmic replacement of endogenous mtDNA [8].

Using the cybrid approach we created a panel of xenomitochondrial cybrid cell lines that contain nuclear DNA from the common laboratory mouse *Mus musculus domesticus* and mtDNA from *Mus musculus domesticus*, or exogenous mtDNA from progressively divergent mouse species *Mus spretus*, *Mus terricolor*, *Mus caroli* and *Mus pahari* [9]. Genetic variation in the mtDNA of these species spans 2–8 million years, and therefore this panel models a wide range of nuclear-mitochondrial genetic divergence that has not been achieved with other research models. Furthermore, we utilized a xenomitochondrial mouse model generated in our laboratory that harbors wild-type, C57BL/6J *Mus musculus domesticus* nuclear genetic background and homoplasmic mtDNA from *Mus terricolor* [10, 11].

Here, we investigated the transcriptomic and functional consequences of nuclear-mtDNA divergence *in vitro* and *in vivo*. RNA sequencing analysis of xenomitochondrial cybrids revealed an activation of interferon signaling pathways and an elevation of secreted interferon alpha under basal conditions, even in the absence of OXPHOS dysfunction. In contrast, xenomitochondrial mice displayed lower baseline interferon gene expression and an impairment in the interferon-dependent innate immune response upon immune challenge. Our work demonstrates that nuclear-mitochondrial genetic divergence that does not result in overt OXPHOS dysfunction can modulate the interferon immune response.

## Materials and methods

### Xenomitochondrial cybrid cell lines

Xenomitochondrial LM-thymidine kinase-negative fibroblast cybrid cell lines harboring nuclear DNA from *Mus musculus domesticus* and matching mtDNA from *Mus musculus domesticus* (Mus^Mus, control; nDNA^mtDNA) or exogenous mtDNA from *Mus Spretus* (Mus^Spretus), *Mus terricolor* (Mus^Terricolor), *Mus caroli* (Mus^Car) or *Mus pahari* (Mus^Pahari) were generated as described previously [9, 12]. To corroborate the identity of mtDNA in xenomitochondrial cybrids, D-loop sequences of each xenomitochondrial construct were compared to

the reference sequence of each rodent (*Mus musculus domesticus* NC_005089.1; *Mus spretus* NC_025952.1; *Mus terricolor* (also known as *Mus dunni*) NC_001665.2) by PCR and Sanger sequencing as described previously [13] using previously published primers [12]. A full mtDNA sequence was generated for *Mus caroli* as it was not previously available (GenBank accession number MK166027).

## Next-generation sequencing of mitochondrial genomes

DNA was extracted using a QIAamp DNA Mini kit (Qiagen, Cat. #51304), with an on-column RNase A treatment to eliminate residual RNA. Five hundred nanograms of isolated genomic DNA underwent enzymatic fragmentation using 3 μl NEBNext dsDNA Fragmentase in a total volume of 20 μl for 30 min, followed by AMPure bead isolation and elution in 20 μl TE buffer. Fragmentation pattern was checked using Multi-NA instrument (Shimadzu, Japan). Remaining 15 μl of fragmented DNA sample underwent library preparation using NEBNext Ultra II DNA Library Prep Kit for Illumina. Barcoded libraries were pooled and underwent sequencing on MiSeq (2 x 76 bp) with version 3 reagents. Skewer (v0.2.2) [14] was used to remove 3' bases with phred quality less than 20. BFC (version r181) was used to correct sequencing errors with a genome size setting of "3g" and otherwise default settings [15]. To assemble mtDNA, ABySS (version 1.9.0) was run with the following parameters; K-mer range: 37 41 45 and coverage threshold: 3 5 10 20. *Mus musculus domesticus* mtDNA was assembled into a single contig with a K-mer of 41 and coverage threshold of 20. *Mus caroli* mtDNA was assembled into two contigs with a K-mer of 37 and coverage threshold of 3, which could be manually merged into one contig. Sequencing of mtDNA from remaining xenomitochondrial cybrid constructs was performed as described previously [16].

The mtDNA sequence of the Mus$^{Mus}$ control cybrid cells was compared to the mtDNA sequence of *Mus musculus domesticus* primary fibroblasts used for the generation of this cybrid and the *Mus musculus domesticus* GenBank reference sequence (NC_005089.1) to verify that they were identical, and to confirm that no nucleotide changes were introduced as a result of the cybridization procedure.

## Cell culture

Xenomitochondrial cybrid cell lines were maintained in RPMI-1640 medium (Cat. #11875–093, Life Technologies) supplemented with 25 mM D-glucose (final concentration), 0.2 mM uridine, 1 mM pyruvate, 10% fetal bovine serum, 100 units/mL penicillin, and 100 μg/mL streptomycin in a humidified incubator (37˚C; 5% $CO_2$). Prior to experimental procedures, cells were incubated in glucose-free RPMI-1640 medium (Cat. #11879–020, Life Technologies) supplemented with 5.5 mM D-glucose, 0.2 mM uridine, 1 mM pyruvate, 10% fetal bovine serum, 100 units/mL penicillin, and 100 μg/mL streptomycin for 48 h in a humidified incubator (37˚C; 5% $CO_2$).

## Animals

All procedures involving animals in this study were approved by the Alfred Medical Research and Education Precinct Ethics Committee (Ethics number E/1822/2018-C) or the University of Melbourne Animal Ethics Committee in strict accordance with Australian governmental regulations. Wild-type C57BL/6J (control) and xenomitochondrial (xeno) mice harboring homoplasmic mtDNA from *Mus terricolor* [10, 11], both male and female, and aged between 6 and 12 weeks old were used (unless specified otherwise). Mice were housed under specific-pathogen-free conditions for HSV-1 infection experiments.

## Cellular proliferation and doubling time determination

Cell proliferation was assessed by counting cells manually using a hemocytometer at 24 h intervals between 1–6 days. The 'exponential growth' non-linear regression model and the fitting method of 'least squares (ordinary fit)' was used to automatically calculate doubling time for each set of replicate counts for each cell line, using Prism 7.0 software. Statistical analysis was conducted using the unpaired, two tailed, Student's $t$-test and the Holm-Sidak method to correct for multiple comparisons.

## High-resolution mitochondrial respiration

Basal, ADP-stimulated complex I, convergent complex I + II and maximal oxygen consumption rates were measured in each xenomitochondrial cybrid cell line using $4 \times 10^6$ cells per chamber in a high-resolution Oroboros Oxygraph 2 K (Oroboros Instruments) as described previously [17]. Briefly, respiration was measured by sequential injection of glutamate (10 mM), malate (2 mM), digitonin (20 μg/ml), ADP (1 mM), succinate (10 mM), oligomycin (2.5 μM), CCCP (1.5 μM), rotenone (1 μM) and antimycin A (2 μM). Residual oxygen consumption (following addition of antimycin A) and leak respiration (after addition of oligomycin) were measured to verify an absence of non-mitochondrial respiration and to determine proton leak, respectively. Measurements were normalized to cell number per chamber and data were analyzed using the Datlab2 software (Oroboros Instruments).

## Measurement of mitochondrial superoxide generation

Cells were incubated with 2 μg/ml Hoechst 33342 (Thermo Fisher Scientific, H1399) and 3 μM MitoSOX (Thermo Fisher Scientific, M36008) in 1x Hank's Balanced Salt Solution (Thermo Fisher) for 20 min at 37°C and 5% $CO_2$. Cells were imaged every 30 min using an ArrayScan Vti High Content Analyzer (Thermo Scientific, MA, USA), with Hoechst 33342 and MitoSOX fluorescence excited with 365 nm and 549 nm laser lines respectively. Fluorescence measurements were averaged from eight wells containing approximately $5 \times 10^4$ cells per well. Significant differences were determined by ANOVA with Tukey's post-hoc multiple comparison tests.

## Blue Native PAGE (BN-PAGE)

40 μg of isolated mitochondria were solubilized for 30 min on ice in 50 μl of 1% (v/v) Triton X-100 (Sigma) or 1% (w/v) digitonin (Merck, NJ, USA). Samples were processed as described previously [18] and resolved on a 4–13% (w/v) BN-PAGE gel at 100 V / 5 mA for approximately 14 h at 4°C. Native proteins were transferred to PVDF membrane using a semi-dry method, blocked with 10% (w/v) skim-milk in 1x PBS / 0.05% (v/v) Tween-20, then probed overnight at 4°C with primary antibodies against OXPHOS complex II (SDHA, Abcam, ab14715), complex IV (COI, Abcam, ab14705), complex I and I/III$_2$ super complex (NDUFA9, raised in rabbits [19]) and complex V (ATP5a, Abcam, ab14748). Immunoblotting membranes were subsequently incubated with the appropriate horseradish peroxidase-coupled secondary antibody and proteins visualized with ECL (GE Healthcare, Little Chalfont, Buckinghamshire, UK) using a Microchemi 4.2 Gel Imaging System (DNR Bio-imaging Systems, Jerusalem, Israel) with a 16-bit CCD camera. Densitometric analyses were conducted on three separate immunoblots, with the band intensities obtained for each Mus$^{Mus}$ control set as 100% protein expression for each OXPHOS complex detected. Protein abundance in each xenomitochondrial cybrid construct is shown relative to this control.

## Fluorescence confocal microscopy

Xenomitochondrial cybrid cell lines were plated onto 13 mm diameter glass coverslips and allowed to attach overnight. Cells were incubated with 100 nM MitoTracker[TM] Red CMXRos (Cat. #M7512, Thermo Fisher Scientific) and Hoechst 33342 (1 μg/ml; Invitrogen) and processed as described previously [20]. Images were acquired using a laser scanning confocal microscope (Nikon A1r) with a 63x/1.40 oil immersion objective. Analysis of mitochondrial network descriptors, aspect-ratio -to measure mitochondrial length, and form factor—a measure of mitochondrial length and branching was conducted using Image J version 1.51p [21].

## RNA sequencing

Total RNA was extracted from three independent biological replicates from each xenomito-chondrial cybrid using the miRNeasy RNA extraction kit (Qiagen, Cat. #217004) with an on-column DNase I (Qiagen) treatment according to manufacturer's instructions. Transcrip-tome-wide mRNA sequencing was performed by the Australian Genome Research Facility (Melbourne, Australia). RNA purity and integrity were confirmed by BioAnalyser (Agilent, CA). Libraries were created with TruSeq RNA v2 kit (Illumina), barcoded and sequenced on a HiSeq 2000 sequencer (Illumina), producing 100 bp single-end reads, with a depth of at least 20 million reads per sample.

The primary sequence data were generated using the Illumina bcl2fastq 1.8.4 pipeline. Reads were trimmed for quality using a minimum phred value of 20 and minimum length of 18 using Skewer version 0.2.2 [14]. Reads were mapped with STAR [22] to the Ensembl mouse reference genome (GRCm38). FeatureCounts [23] was used to count reads mapped to exons of genes with a minimum mapping quality of 20 using Ensembl genome annotation (Mus musculus.GRCm38.77.gtf) to generate a count matrix. Genes with an average of 10 reads or less per sample across the experiment were excluded from downstream analysis.

**RNA sequencing differential gene expression.** To quantify changes in gene expression between xenocybrid cell lines, the Mus[Mus] sample was set as baseline. The count matrix under-went differential analysis with the EdgeR package [24] comparing Mus[Mus] control to all other xenomitochondrial cybrid sample groups and heatmaps of scaled gene-wise count data were generated in R. Pathway analysis was conducted with Pre-Ranked Gene Set Enrichment Analysis [25] using curated gene sets from Reactome [26]. Genes were ranked from most up-regulated to most down-regulated by multiplying the sign of the log2 fold change by the inverse of the *p* value as described previously [27]. To visualize pathway regulation across several contrasts while retaining inter-sample variance information, we used a differential rank sum approach we call "gsheat". Differential rank-sum analysis was performed using Limma to prioritize gene sets with the smallest *p* values across the contrasts. Gsheat code has been deposited to Github (https://github.com/markziemann/gsheat).

## qRT-PCR

RNA was isolated using the miRNeasy RNA extraction kit (Qiagen, Cat. #217004) with an on-column DNase I (Qiagen) treatment to eliminate contaminating genomic DNA, according to manufacturer's instructions, and cDNA was prepared using the QuantiTect Reverse Transcription Kit (Qiagen 205311). qRT-PCR was performed as described previously [28] using a StepOnePlus Real-Time PCR System (Thermo Fisher Scientific) and TaqMan gene expression assays: *Ifi44* Mm00505670_m1; *Isg15* Mm01705338_s1; *Irf7* Mm00516793_g; *Stat2* Mm00490880_m1; *Actb* 4352341E. Relative quantitation (fold RNA change) was obtained by applying the comparative $C_T$ method [29] and normalized against the level of the reference gene β-Actin (Actb).

## ELISA

IFNα was detected using the IFNα mouse ELISA kit (Thermo Fisher Scientific, Cat# BMS6027) according to manufacturer's instructions.

## HSV-1 infection and titer determination

Age-matched (6–12 weeks old), wild-type and xeno female mice were infected with $10^6$ plaque-forming units (PFU) of HSV-1 strain KOS on the skin by flank scarification [30]. Briefly, the left flank of anesthetized mice was shaved, and remaining hair removed using commercially available depilation cream. A small area of skin was then abraded for virus inoculation and mice were bandaged for 48 h. Infectious virus from 0.5 x 2-cm pieces of full-thickness skin from the site of infection was measured by the standard assay of PFU as previously described [30]. The HSV-1 KOS strain was propagated and titrated using Vero cells (CSL, Parkville, Australia).

## Statistical analysis

Results are presented as the mean ± SD or mean ± SEM as indicated on the legend. Statistical analysis was performed with Prism 5.01 software (GraphPad Software Inc.) using two-tailed, unpaired Student's $t$-test unless otherwise indicated. Exact $p$ values are indicated in the legend, a $p$ value of $<0.05$ was considered as significant.

# Results

## Generation of mouse xenomitochondrial cybrids

To investigate the nuclear genetic response to divergent mitochondrial genotypes we generated five xenomitochondrial mouse fibroblast cybrid lines harboring nuclear DNA (nDNA) from *Mus musculus domesticus* and mtDNA from *Mus musculus domesticus* (Mus$^{Mus}$, control; nDNA$^{mtDNA}$) or exogenous mtDNA from progressively divergent rodent species *Mus spretus* (Mus$^{Spretus}$), *Mus terricolor* (Mus$^{Terricolor}$), *Mus caroli* (Mus$^{Caroli}$) or *Mus pahari* (Mus$^{Pahari}$) (S1A Fig). The genetic divergence amongst the mtDNA donors relative to *Mus musculus domesticus* was determined by whole mtDNA next-generation sequencing (Table 1). This analysis confirmed an increasing nuclear-mitochondrial genetic divergence as a result of the introduction of exogenous mtDNA into a common *Mus musculus domesticus* nuclear genetic background (S1B Fig).

## Divergent mtDNA genotypes decrease OXPHOS protein levels but do not compromise bioenergetic function

As mtDNA encodes genes for core OXPHOS subunits, we measured mitochondrial respiration to determine the functional consequences of divergent mtDNA in xenomitochondrial cybrids with *Mus musculus* nuclear background. Basal and maximal respiration were measured in permeabilized cells (Fig 1A). Unexpectedly, significant increases in basal and maximal respiratory rates were detected in the closest-related cybrid Mus$^{Spretus}$ relative to Mus$^{Mus}$ control, while no changes were detected in Mus$^{Terricolor}$ or Mus$^{Caroli}$ cybrids (Fig 1A). A trend for

**Table 1. mtDNA amino-acid differences between xenomitochondrial cybrid constructs.**

| Species | GenBank number | Amino-acid similarity (%) |
|---|---|---|
| *Mus musculus* | KY018919 | 100 |
| *Mus spretus* | KY018921 | 93.16 |
| *Mus terricolor* | KY018920 | 89.14 |
| *Mus caroli* | MK166027 | 86.77 |
| *Mus pahari* | KY038052 | 85.4 |

decreased maximal respiration was observed in the most divergent cybrid Mus$^{Pahari}$ (73% of Mus$^{Mus}$ control) but it did not reach statistical significance. OXPHOS complex protein levels were analyzed by blue-native PAGE (BN-PAGE; Fig 1B and S2 Fig). Decreased complex I and V levels were detected in Mus$^{Caroli}$ and Mus$^{Pahari}$ cybrids, while complex IV levels were decreased in the most divergent cybrid Mus$^{Pahari}$ relative to control (Fig 1B and S2 Fig). Complex III protein abundance was decreased in Mus$^{Caroli}$ relative to control (S2 Fig).

Additional parameters of cell and mitochondrial health were assessed in xenomitochondrial cybrids. Doubling time and mitochondrial superoxide production were similar amongst all cybrids (Fig 1C and 1D). Furthermore, an assessment of cellular morphology and mitochondrial network structure by confocal microscopy, using MitoTracker$^{TM}$ Red and Hoechst 33342 to stain mitochondria and nuclei respectively, did not reveal overall morphological changes

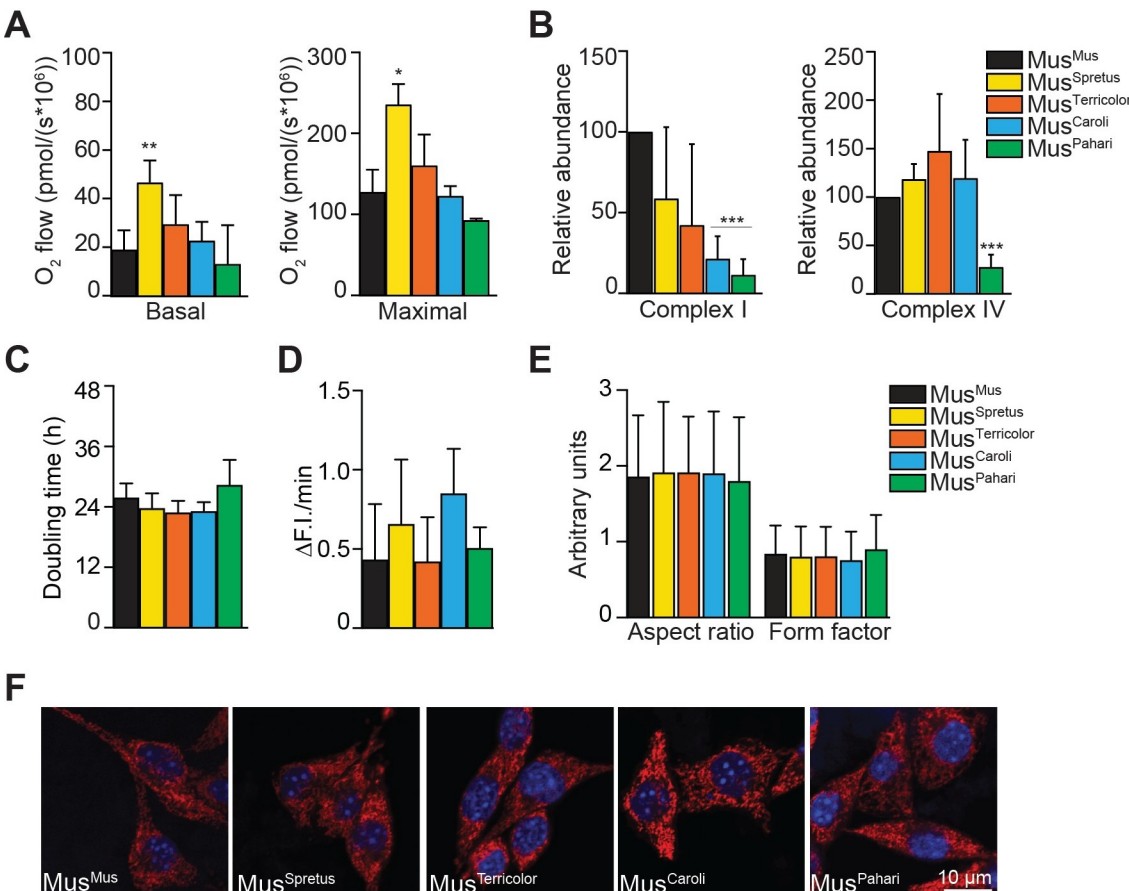

**Fig 1. Divergent mtDNA genotypes decrease OXPHOS protein levels but do not compromise bioenergetic function.** (A) Oxygen consumption rates were measured in digitonin-permeabilized xenomitochondrial cybrids. Basal and maximal respiration rates are increased in Mus$^{Spretus}$ compared with Mus$^{Mus}$ control ($p = 0.009$ and $0.031$ respectively). Data presented as mean ± SD (n ≥ 3); $^{*}p < 0.05$ and $^{**}p < 0.01$ by unpaired, two tailed, Student's $t$-test. (B) OXPHOS complexes protein levels in xenomitochondrial cybrids. Densitometric analysis of BN-PAGE immunoblots showing a decrease in complex I protein abundance in Mus$^{Caroli}$ and Mus$^{Pahari}$ ($p < 0.001$ for both samples) and reduced complex IV levels in Mus$^{Pahari}$ ($p < 0.001$) relative to Mus$^{Mus}$ control. Protein abundance was normalized to nuclear-encoded complex II expression levels. Data presented as mean ± SD (n = 3); $^{**}p < 0.01$ by unpaired, two tailed, Student's $t$-test. (C) Doubling time is similar amongst xenomitochondrial cybrids. (D) Superoxide production rates were similar in xenomitochondrial cybrids. Rates were determined by changes in fluorescence intensity (ΔF.I.) using MitoSOX over a 30 min period. Data presented as mean ± SEM (n = 3). (E) Mitochondrial network aspect-ratio and form factor are unchanged in xenomitochondrial cybrids. (F) Representative images of xenomitochondrial cybrid cells stained with MitoTracker$^{TM}$ Red CMXRos (mitochondria; red) and Hoechst 33342 (nucleus; blue). Images were acquired using a laser scanning confocal microscope (Nikon A1r) with a 63x/1.40 oil immersion objective. Scale bar is 10 μm.

(Fig 1E and 1F). Taken together, these results indicate that the presence of divergent mtDNA does not compromise overall cellular health or mitochondrial bioenergetic function, despite significant decreases in OXPHOS complex protein levels in highly divergent cybrids.

## Divergent mtDNA genotypes induce interferon pathway gene expression

Next, we used RNA sequencing to elucidate the transcriptomic changes associated with nuclear-mtDNA divergence in xenomitochondrial cybrids. We used three biological replicates per xeno-mitochondrial cybrid and generated 20 million reads per sample. Differentially-expressed genes corrected for false discovery rate (FDR < 0.05) ranged between 3,364 in Mus$^{Terricolor}$ to 6,147 in Mus$^{Pahari}$ relative to Mus$^{Mus}$ control (Table 2). Notably, despite Mus$^{Spretus}$ having the most closely related mtDNA to Mus$^{Mus}$, gene expression in this cybrid was markedly different to control cybrid, as we detected 4,748 genes differentially expressed between these two constructs.

Gene set enrichment analysis revealed an enrichment of the interferon signaling pathway, including interferon alpha, beta, gamma and cytokine signaling in xenomitochondrial cybrids relative to Mus$^{Mus}$ control cybrid (Fig 2A). Enrichment of this pathway was highest in Mus$^{Spretus}$ and Mus$^{Terricolor}$ cybrids. The most highly expressed interferon-stimulated genes (ISGs) included genes with direct antiviral activity (*Ifi44*, *Isg15*, *Ifit1*, *Ifit3*, *Oas*, *Rtp4*), cytoplasmic DNA and RNA sensors (*Ddx58*, *Ifi203*, *Ifi204*) and transcription factors that enhance the anti-viral response (*Irf7*, *Stat1* and *Stat2*) (Fig 2A). These results were corroborated in the cybrids with the highest interferon signal (Mus$^{Spretus}$ and Mus$^{Terricolor}$) using three representative ISGs (*Ifi44*, *Isg15 and Irf7*) by qRT-PCR (Fig 2B). Interferon alpha (IFNα) production was also increased in Mus$^{Spretus}$ and Mus$^{Terricolor}$ cybrids compared to control cybrid, as measured by ELISA (Fig 2C), indicating that increased levels of ISGs also results in higher production of IFNα in these cells. Together, these results are consistent with an enhanced type I interferon cellular response in xenomitochondrial cybrid cells.

## Divergent mtDNA suppresses type I interferon activation *in vivo*

To investigate the biological significance of nuclear-mtDNA divergence *in vivo*, we utilized the mouse version of the xenomitochondrial cybrid Mus$^{Terricolor}$. Although Mus$^{Spretus}$ showed the highest activation of the interferon pathway, we did not succeed in producing the correspond-ing xeno mouse line [10]. Xenomitochondrial "xeno" mice harbor wild-type, C57BL/6J *Mus musculus domesticus* nuclear DNA and homoplasmic mtDNA from *Mus terricolor* [10, 11]. An initial characterization of the xeno mouse had not revealed an overt phenotype in animals aged 14 months-old or younger [31].

Given the new insights we obtained by RNA sequencing, indicating an activation of ISGs in xenomitochondrial cybrids *in vitro*, we measured steady-state levels of representative ISGs (*Ifi44*, *Irf7*, *Isg15*, *Stat2*) in the kidney and the liver (Fig 3A and 3B respectively) from 3-month-old, sex-matched xeno and wild-type C57BL/6J ("control") mice by qRT-PCR. Unex-pectedly and contrary to our *in vitro* results, all ISGs measured were down-regulated in the kidney from xeno mice relative to control mice (Fig 3A). Additionally, *Ifi44* and *Isg15* were

**Table 2. Number of differentially-expressed genes (FDR<0.05) in the up and down direction in each xenomito-chondrial cybrid cell line against Mus$^{Mus}$ control as determined by edgeR.**

| Contrast | Up | Down | Up + Down |
|---|---|---|---|
| Mus$^{Mus}$—Mus$^{Terricolor}$ | 1924 | 1440 | 3364 |
| MusMus—Mus$^{Caroli}$ | 2327 | 2081 | 4408 |
| MusMus—Mus$^{Spretus}$ | 2539 | 2209 | 4748 |
| MusMus—Mus$^{Pahari}$ | 2982 | 3165 | 6147 |

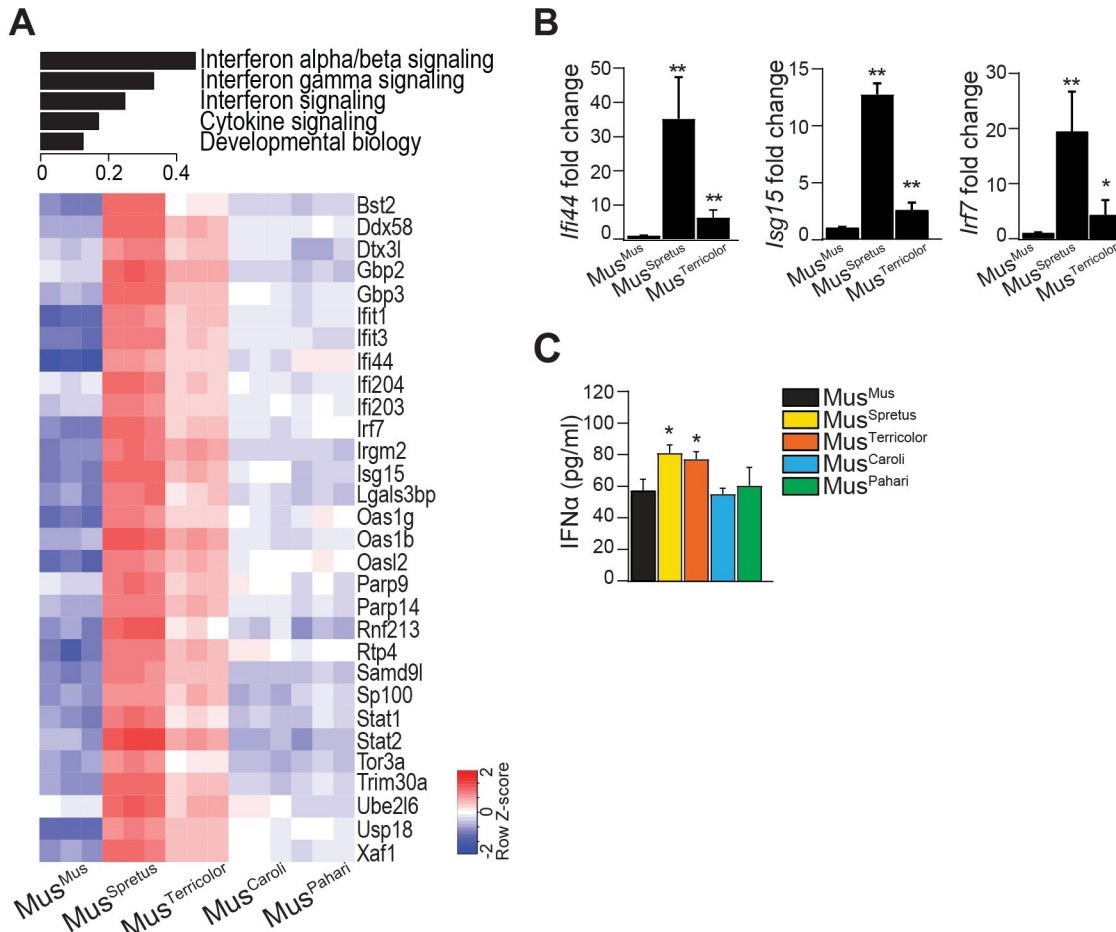

**Fig 2. Divergent mtDNA genotypes induce interferon pathway gene expression.** (A) Pathway analysis from RNA sequencing using Reactome gene sets across xenomitochondrial cybrids relative to Mus[Mus] control. Interferon signaling pathways were the most differentially expressed pathways. GSEA-P FDR<0.05 for all gene sets shown. Heatmap shows expression of ISGs in xenomitochondrial cybrids relative to Mus[Mus] control. (B) qRT-PCR analysis of representative ISGs *Ifi44*, *Isg15* and *Irf7*. Relative gene expression is up-regulated in Mus[Spretus] and Mus[Terricolor] xenomitochondrial cybrids relative to Mus[Mus] (*Ifi44 p* = 0.002, 0.003; *Isg15 p* <0.001, 0.003; *Irf7 p* = 0.003, 0.048 respectively). Gene expression was normalized to *Actb* levels. (C) Interferon alpha (IFNα) secretion is increased in supernatant from Mus[Spretus] and Mus[Terricolor] cybrids relative to Mus[Mus] (*p* = 0.006 and 0.014 respectively) as determined by ELISA. Data for B and C are presented as mean ± SD (n ≥ 3); *p* < 0.05 and **p* < 0.01 by unpaired, two tailed, Student's *t*-test.

decreased in the liver from xeno mice (Fig 3B). Taken together, this suggests a lower baseline ISG expression in xeno mice.

Interferons act as a critical first line of defense against viral infections [32, 33]. Therefore, we challenged xeno mice with herpes simplex virus type 1 (HSV-1), as the immunological control of this virus is strongly dependent on type I interferons [34, 35]. Mice were infected epicutaneously with HSV-1, as this mimics the natural route of infection during which the virus can undergo rounds of local replication in the skin [30]. Type I interferons are secreted within hours at the site of infection and can be detected for several days following local HSV infection [36]. Adaptive immunity, such as T cells that are involved in viral clearance, is initiated from day 2 onwards in the skin-draining lymph nodes [30, 37]. The immune system of wild-type mice generally can clear the virus from the skin within 7 to 8 days, whereas mice unable to respond to type I interferons show increased viral titers, spread of the virus to the central nervous system and succumb to the infection [38, 39].

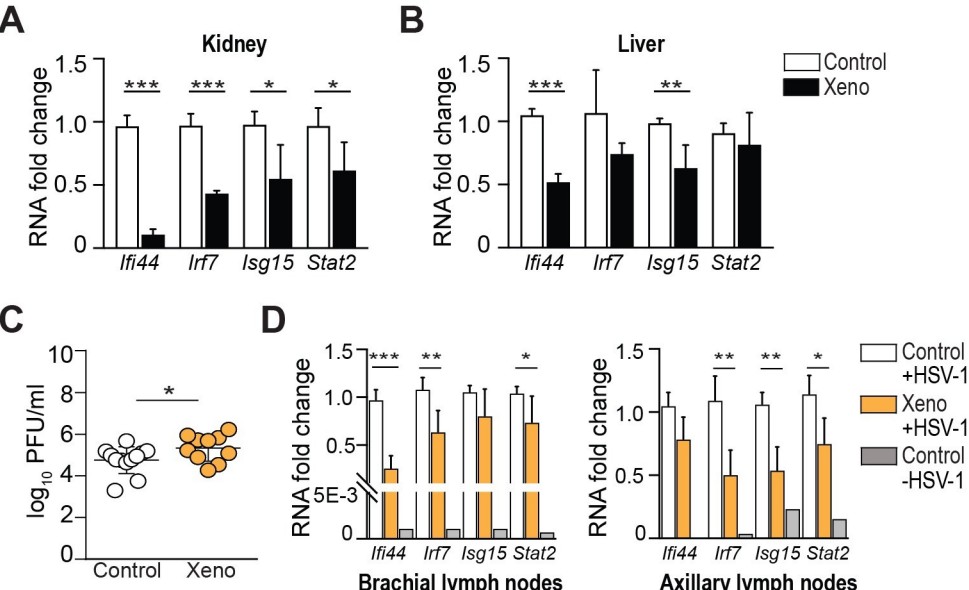

**Fig 3. Divergent mtDNA suppresses type I interferon activation *in vivo*.** Steady-state levels of representative ISGs (*Ifi44*, *Irf7*, *Isg15*, *Stat2*) in the kidney (A) and the liver (B) from 3-month-old, sex-matched wild-type C57BL/6J ("control") and Mus<sup>Terricolor</sup> xenomitochondrial ("xeno") mice by qRT-PCR. All ISGs measured were down-regulated in the kidney from xeno mice relative to control mice (*Ifi44* $p < 0.001$, *Irf7* $p < 0.001$, *Isg15* $p = 0.025$ and *Stat2* $p = 0.036$). Additionally, *Ifi44* and *Isg15* were decreased in the liver from xeno mice ($p < 0.001$ and $p = 0.003$ respectively). Data are presented as mean ± SD, n ≥ 4 mice per experimental group; *$p < 0.05$, **$p < 0.01$ and ***$p < 0.001$ by unpaired, two tailed, Student's *t*-test. (C) Mice were infected on the flank skin with HSV-1 and viral titers were measured 2 days post-infection. Viral titers in xeno mice were increased compared to control mice. Data are presented as mean ± SD, n = 12 mice (control) and n = 10 mice (xeno); $p = 0.045$ by unpaired, two tailed, Student's *t*-test. (D) ISG expression in the brachial and axillary lymph nodes following HSV-1 infection. *Ifi44*, *Irf7* and *Stat2* mRNA levels were decreased in brachial lymph nodes of xeno mice 2 days post-HSV1 infection (*Ifi44* $p < 0.001$, *Irf7* $p = 0.009$ and *Stat2* $p = 0.048$). *Irf7*, *Isg15* and *Stat2* mRNA levels were decreased in axillary lymph nodes 7 days post-infection (*Irf7* $p = 0.005$, *Isg15* $p = 0.002$ and Stat2 $p = 0.019$). ISG expression was measured in one naïve control mouse (Control—HSV-1) to confirm interferon activation upon infection. Data are presented as mean ± SD, n ≥ 4 mice per experimental group; *$p < 0.05$, **$p < 0.01$ by unpaired, two tailed, Student's *t*-test.

Similar to control animals, xeno mice did not show severe signs of disease following epicutaneous HSV-1 infection. However, viral titers in xeno mice were significantly increased compared to control mice on day 2 post-infection, suggesting a decreased innate immune response (Fig 3C). To further investigate this phenomenon, we analyzed ISG expression of the skin-draining lymph nodes following HSV-1 infection. As expected, ISG expression was increased following HSV-1 infection of control mice relative to naïve control (Fig 3D). Interestingly, we observed a decrease in ISG expression in brachial lymph nodes of xeno mice 2 days following infection, with *Ifi44*, *Irf7*, and *Stat2* mRNA levels reduced. Similarly, *Irf7*, *Isg15* and *Stat2* mRNA levels were decreased in axillary lymph nodes of xeno mice 7 days post-infection relative to control mice (Fig 3D). Collectively, these results indicate that nuclear-mtDNA divergence in xeno mice results in lower baseline ISG expression, and a subsequent impairment in the interferon-mediated innate immune response upon epicutaneous infection with HSV-1, which results in limited viral control.

## Discussion

Nuclear-mitochondrial genetic communication is an intrinsic component of cellular function. Here we show that nuclear-mitochondrial genetic divergence due to the introduction of

exogenous mtDNA can result in major transcriptomic and functional changes both *in vitro* and *in vivo*, even when OXPHOS function is not compromised.

Xenomitochondrial cybrids did not display impaired OXPHOS function, despite significant decreases in OXPHOS complex protein abundance. Notably, their transcriptomic profile revealed an activation of interferon signaling pathways even in the absence of immune challenge. A similar activation of the type I interferon pathway has been described in pathogenic contexts. For instance, in response to mtDNA instability due to deficiency in mitochondrial transcription factor a (TFAM) [40], mtDNA depletion [41] or as a consequence of increased levels of unstable mitochondrial double-stranded RNA species [42]. Our work demonstrates that the interferon response can be activated independently of genetic mutations or manipulations that impair directly the stability of mitochondrial DNA or RNA.

Our work indicates that nuclear-mitochondrial genetic divergence that does not result in overt OXPHOS dysfunction can modulate the interferon immune response, although the mechanism remains to be elucidated. Contrary to our *in vitro* results, xeno mice displayed lower baseline ISG expression, and an impairment in the interferon-dependent innate immune response upon immune challenge with HSV-1, which resulted in decreased viral control. The L-cell nucleus of the cybrids is aneuploid, with a modal number of 45 chromosomes [43], which we speculate may influence the nuclear response observed *in vitro* compared to the euploid nucleus of xeno mouse cells. Our results show that the interferon-dependent innate immune response is susceptible to modulation by subtle differences in OXPHOS consequent to nuclear-mtDNA divergence, and that the direction of this modulation may also depend on the nuclear background. Additional work is required to understand how the interferon response may change in different tissues of the xeno mouse when aging or other relevant stressors are imposed.

Studies in conplastic mouse strains harboring the nuclear genome of C57BL/6 mice and mtDNA of wild-type NZB/OlaHsd mice have shown effects on metabolic function and aging [5], whereas mitochondrial-nuclear exchange (MNX) mice that harbor mtDNA from the C3H/HeN strain and the nuclear genome of the C57BL/6 strain display higher reactive oxygen species associated with susceptibility to pathological changes in the heart [6]. These examples illustrate that nuclear-mitochondrial genetic divergence can result in significant phenotypic effects, ranging from enhanced metabolic fitness to pathological changes. Our work adds the interferon response as a new pathway warranting investigation in such models.

There is growing evidence that patients affected by mitochondrial disease experience immune dysfunction and suffer from recurrent infections which can be life-threatening [44]. However, current mouse models of primary mitochondrial disease may not recapitulate this emerging aspect of mitochondrial disease in humans. The xeno mouse may provide a useful model to investigate the immune response upon additional mitochondrial challenges, such as aging.

The pathogenesis of mitochondrial diseases is traditionally ascribed to bioenergetic failure or oxidative stress. By revealing an unexpected link between mtDNA genotype and the interferon response, our work offers a new perspective on mitochondrial diseases where an increased susceptibility to viral infection is well known. Further work in models of both non-pathogenic and disease-causing mtDNA variants is warranted to investigate potential effects on the interferon immune response, as we speculate that nuclear-mitochondrial interactions may contribute to inter-individual differences seen in immune response between healthy humans.

## Supporting information

**S1 Fig. Generation of mouse xenomitochondrial cybrids.** (A) Xenomitochondrial LM-thymidine kinase-negative fibroblast cybrid lines were generated by fusion of *Mus musculus*

*domesticus* (Mus musculus) cells chemically devoid of mtDNA with enucleated cells harboring mtDNA from *Mus musculus domesticus* (Mus<sup>Mus</sup>; control) or exogenous mtDNA from *Mus spretus* (Mus<sup>Spretus</sup>), *Mus terricolor* (Mus<sup>Terricolor</sup>), *Mus caroli* (Mus<sup>Caroli</sup>) and *Mus pahari* (Mus<sup>Pahari</sup>) to model progressively-increasing nuclear-mitochondrial genetic divergence. (B) Maximum parsimony phylogenetic tree generated from whole mtDNA sequences using Clustal Omega [45] and NJPlot software [46] show the genetic relationship between xenomitochondrial cybrid constructs. Bootstrap values are shown above the nodes (%, from 10,000 trials). Scale bar indicates percentage of substitutions per site. The tree is rooted using *Lepus granatensis* mtDNA sequence data (accession number NC_024042.1).
(TIF)

**S2 Fig. OXPHOS complexes protein abundance in xenomitochondrial cybrids by BN-PAGE.** Densitometric analysis of OXPHOS complexes III (A) and V (B) by BN-PAGE show a decrease in complex III protein levels in Mus<sup>Caroli</sup> and reduced complex V levels in Mus<sup>Caroli</sup> and Mus<sup>Pahari</sup> relative to Mus<sup>Mus</sup> control. Protein abundance was normalized to nuclear-encoded complex II expression levels. Data presented as mean ± SD (n = 3). (C) Representative BN-PAGE immunoblot.
(TIF)

**S1 Table. Data used to generate high-resolution mitochondrial respiration graphs showing mean ± standard deviation.**
(DOCX)

**S2 Table. Quantification of densitometric analysis from BN-PAGE immunoblots showing mean ± standard deviation relative to Mus<sup>Mus</sup> control (set as 100% protein expression).**
(DOCX)

**S3 Table. Data used to generate doubling time graph showing mean ± standard deviation.**
(DOCX)

**S4 Table. Data used to generate superoxide production graph showing mean ± SEM.**
(DOCX)

**S5 Table. Data used to generate mitochondrial network aspect-ratio and form factor graph showing mean ± standard deviation.**
(DOCX)

**S6 Table. Data used to generate graph showing log2 RNA fold change of representative ISGs showing mean ± standard deviation.**
(DOCX)

**S7 Table. ELISA data used to generate IFNα production graph showing mean ± standard deviation.**
(DOCX)

**S8 Table. Data used to generate graph showing log2 RNA fold change of representative basal ISG expression in kidney and liver showing mean ± standard deviation.**
(DOCX)

**S9 Table. Data used to generate viral titers graph showing mean ± standard deviation.**
(DOCX)

**S10 Table. Data used to generate graph showing log2 RNA fold change of representative ISGs in brachial and axillary lymph nodes following HSV-1 infection, showing**

**mean ± standard deviation.**
(DOCX)

**S1 Raw images.**
(PDF)

# Acknowledgments

We thank and acknowledge the Australian Genome Research Facility (AGRF) for high throughput sequencing, Sheridan Keene for providing valuable technical assistance and Stephanie Lopez for assistance with figure preparation.

# Author Contributions

**Conceptualization:** M. Isabel G. Lopez Sanchez, Ian A. Trounce.

**Data curation:** Mark Ziemann, Annabell Bachem, Rahul Makam, Matthew McKenzie.

**Formal analysis:** M. Isabel G. Lopez Sanchez, Mark Ziemann, Annabell Bachem, Rahul Makam, Matthew McKenzie.

**Funding acquisition:** Jonathan G. Crowston, Ian A. Trounce.

**Investigation:** M. Isabel G. Lopez Sanchez, Mark Ziemann, Matthew McKenzie.

**Methodology:** M. Isabel G. Lopez Sanchez, Mark Ziemann, Annabell Bachem, Sammy Bedoui.

**Project administration:** M. Isabel G. Lopez Sanchez.

**Resources:** Carl A. Pinkert, Sammy Bedoui, Ian A. Trounce.

**Supervision:** M. Isabel G. Lopez Sanchez, Ian A. Trounce.

**Validation:** M. Isabel G. Lopez Sanchez.

**Writing – original draft:** M. Isabel G. Lopez Sanchez, Annabell Bachem, Matthew McKenzie, Sammy Bedoui, Ian A. Trounce.

**Writing – review & editing:** M. Isabel G. Lopez Sanchez, Mark Ziemann, Annabell Bachem, Jonathan G. Crowston, Carl A. Pinkert, Matthew McKenzie, Sammy Bedoui, Ian A. Trounce.

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
