## [Decision Letter · Decision Letter 0]

18 Jun 2020

PONE-D-20-16529

Nuclear response to divergent mitochondrial DNA genotypes modulates the interferon immune response

PLOS ONE

Dear Dr. Lopez Sanchez,

Thank you for submitting your manuscript to PLOS ONE. After careful consideration, we feel that it has merit but does not fully meet PLOS ONE’s publication criteria as it currently stands. Therefore, we invite you to submit a revised version of the manuscript that addresses the points raised during the review process.

We look forward to receiving your revised manuscript.

Kind regards,

Yidong Bai

Academic Editor

PLOS ONE

Journal Requirements:

2. Thank you for including the following funding statement within your acknowledgements section; "We acknowledge the financial support provided by the Mito Foundation (M.I.G.L.S) and National Health and Medical Research Council (I.A.T.). CERA receives Operational Infrastructure Support from the Victorian Government"

"The authors received no specific funding for this work."

Reviewers' comments:

Reviewer's Responses to Questions

**Comments to the Author**

1. Is the manuscript technically sound, and do the data support the conclusions?

Reviewer #1: Yes

2. Has the statistical analysis been performed appropriately and rigorously? 

Reviewer #1: Yes

3. Have the authors made all data underlying the findings in their manuscript fully available?

Reviewer #1: No

4. Is the manuscript presented in an intelligible fashion and written in standard English?

Reviewer #1: Yes

5. Review Comments to the Author

Reviewer #1: This paper addresses an important question related to mito-nuclear interaction, and the functional consequences not only for mitochondrial function but also for nuclear gene expression, cellular function, and organismal responses to viral immune challenge. The data is well presented also some elements could be improved, and the conclusions appear robust. The use of both in vitro and in vivo approaches provide interesting diverging results that future research will have to resolve. Based on differences in cellular metabolic rates between in vivo and in vitro conditions, this finding may not be surprizing and could reflect the fact that immune responses in vivo are the product of collective action of processes across cell types and organ systems, whereas the in vitro situation with cell monocultures is a vastly different system where each cell is responsible for itself. Below are some comments to help improve the manuscript.

• The authors state that all cybrid cell lines have “identical nuclear background”. However, cybrids often have unstable nuclear genomes and have abnormal karyotype, as stated in the discussion. Was this specifically examined to confirm the identical nature of the cell final cell lines generated at the time of study?

• Lines 99-100: how large is the trend? The use of effect sizes (Cohen’s d or Hedges’s g) or at least a % difference would be useful to the reader. Because the p value is heavily influenced by the sample size, the p value says little about the magnitude and biological significance of differences. Generating (and plotting) effect sizes would allow the authors to make useful comparison about the magnitude of effects between the mtDNAs from difference origins.

• For the blue native protein abundance data, the graphs state “relative expression” – this would be appropriate for gene expression, but here this is protein abundance relative to Complex II. For clarity, this should be specified on the graph “Abundance relative to CII”. The authors should also specify whether these are assembled monomers of the complexes, show some of the BN page gels, and it would be useful to show the results for other complexes – even if in supplemental material.

• Figure 1 E would be more useful if the points were made 50-80% transparent so that all groups can be visualized. Otherwise here it just looks like a big black blob.

• Line 139: should read “interferon-stimulated”.

• Why did the authors use Mus terricolor instead of Mus spretus as the xenomitochonrial model? In cybrids, Mus spretus showed the most activation of IFN pathway. Is it because Mus terricolor is relatively divergent?

• Could you comment on the kinetics of IFNs upon infection, is there a peak window post-viral infection that may have been missed while assessing the IFN expression or peripheral response? Providing this information would enhance the impact of this paper and help guide subsequent research on this topic.

• Could age be a factor in the observed IFN response along with genetic background? What was the cytokine or IFN profile-like in the older mice (line 163) compared to the few weeks- old mice used in the study (line 288)?

• Were there changes in mtDNA copy number in the model systems used relative to the wild-type strains, at baseline and post viral challenge?

• Can the authors speculate on the potential translatability of these findings to patients with mitochondrial disease or mitochondrial replacement therapy? There are large inter-individual differences in immune responses in healthy individuals. To what extent do the author think mtDNA-nDNA interactions may account for inter-individual differences in healthy humans?

6. PLOS authors have the option to publish the peer review history of their article (what does this mean?). If published, this will include your full peer review and any attached files.

Reviewer #1: Yes: Kalpita Karan, Martin Picard

---

## [Author Response · Author response to Decision Letter 0]

7 Jul 2020

Dear Editor,

We thank the reviewers for their feedback on our manuscript "Nuclear response to divergent mitochondrial DNA genotypes modulates the interferon immune response". We have now responded to the comments raised by the reviewers and made the relevant changes in our resubmitted manuscript using Track Changes. We believe these changes have substantially improved our manuscript and hope it is now suitable for publication in PLOS ONE.

Editorial requirements 

We have revised the manuscript to make sure it meets PLOS ONE’s style requirements. Figure captions and tables have been inserted immediately after the first paragraph in which they are cited. We have removed figures from the main text and uploaded individual files separately and have renamed “Fig x” instead of “Figure x”. We have also included a supplemental file with raw images for our BN-PAGE immunoblot.

2. Thank you for including the following funding statement within your acknowledgements section; "We acknowledge the financial support provided by the Mito Foundation (M.I.G.L.S) and National Health and Medical Research Council (I.A.T.). CERA receives Operational Infrastructure Support from the Victorian Government"

Please remove any funding-related text from the manuscript and let us know how you would like to update your Funding Statement. Currently, your Funding Statement reads as follows: "The authors received no specific funding for this work."

We have removed the funding-related text from the manuscript and included the relevant Funding Statement online.

Comments to the Author

1. Is the manuscript technically sound, and do the data support the conclusions?

Reviewer #1: Yes

We thank the reviewers for their positive view of our review.

2. Has the statistical analysis been performed appropriately and rigorously? 

Reviewer #1: Yes

 We thank the reviewers for their positive view of our review.

3. Have the authors made all data underlying the findings in their manuscript fully available?

The https://journals.plos.org/plosone/s/editorial-and-publishing-policies requires authors to make all data underlying the findings described in their manuscript fully available without restriction, with rare exception (please refer to the Data Availability Statement in the manuscript PDF file). The data should be provided as part of the manuscript or its supporting information, or deposited to a public repository. For example, in addition to summary statistics, the data points behind means, medians and variance measures should be available. If there are restrictions on publicly sharing data—e.g. participant privacy or use of data from a third party—those must be specified.

Reviewer #1: No

The RNAsequencing datasets generated and analyzed in our study are available in the GEO repository, under accession number GSE113340, and the code for the differential rank-sum analysis approach used to visualize pathway regulation “gsheat” has been deposited to Github (https://github.com/markziemann/gsheat). In addition to summary statistics presented in the manuscript for clarity, we have now included descriptive statistics including means and variance 

measures for all the experiments included in the manuscript (new Supporting File 1). We have also included raw images for the BN-PAGE blots included in our manuscript (S2_raw_images file).

4. Is the manuscript presented in an intelligible fashion and written in standard English?

Reviewer #1: Yes

We thank the reviewers for their positive view of our review.

5. Review Comments to the Author

Reviewer #1: This paper addresses an important question related to mito-nuclear interaction, and the functional consequences not only for mitochondrial function but also for nuclear gene expression, cellular function, and organismal responses to viral immune challenge. The data is well presented also some elements could be improved, and the conclusions appear robust. The use of both in vitro and in vivo approaches provide interesting diverging results that future research will have to resolve. Based on differences in cellular metabolic rates between in vivo and in vitro conditions, this finding may not be surprizing and could reflect the fact that immune responses in vivo are the product of collective action of processes across cell types and organ systems, whereas the in vitro situation with cell monocultures is a vastly different system where each cell is responsible for itself. Below are some comments to help improve the manuscript.

We thank the reviewers for their useful and positive feedback. We have addressed each point below and believe these changes have improved the clarity and quality of our manuscript.

• The authors state that all cybrid cell lines have “identical nuclear background”. However, cybrids often have unstable nuclear genomes and have abnormal karyotype, as stated in the discussion. Was this specifically examined to confirm the identical nature of the cell final cell lines generated at the time of study?

The possibility of minor nuclear differences between clones has not been excluded. We originally produced three independent clones of each cybrid construct, and in earlier work (McKenzie et al 2003 as cited in the manuscript) we found that all 3 clones from each construct gave consistent results in OXPHOS analysis. However, for clarity, we have now removed “identical” to describe the nuclear 

background of the cybrids in the text.

• Lines 99-100: how large is the trend? The use of effect sizes (Cohen’s d or Hedges’s g) or at least a % difference would be useful to the reader. Because the p value is heavily influenced by the sample size, the p value says little about the magnitude and biological significance of differences. Generating (and plotting) effect sizes would allow the authors to make useful comparison about the magnitude of effects between the mtDNAs from difference origins.

The reviewer refers to our description of mitochondrial respiration results in the sentence “A trend for decreased maximal respiration was observed in the most divergent cybrid MusPahari but it did not reach statistical significance.” We have now included descriptive statistics (new Supporting File 1) that enable a direct comparison of the results obtained. We have also modified the sentence to include a % difference as suggested by the reviewer: “A trend for decreased maximal respiration was observed in the most divergent cybrid MusPahari (73% of MusMus control) but it did not reach statistical significance.”

• For the blue native protein abundance data, the graphs state “relative expression” – this would be appropriate for gene expression, but here this is protein abundance relative to Complex II. For clarity, this should be specified on the graph “Abundance relative to CII”. 

We thank the reviewers for this feedback. We have replaced “relative expression” with “relative abundance” on the relevant graphs. The Methods section and legend indicate “Protein abundance in each xenomitochondrial cybrid construct is shown relative to this control.” We have also clarified in the legend “Protein abundance was normalized to nuclear-encoded complex II expression levels.”

The authors should also specify whether these are assembled monomers of the complexes, show some of the BN page gels, and it would be useful to show the results for other complexes – even if in supplemental material.

We have now included raw images of the BN-PAGE blots in our manuscript (S2_raw_images file) showing monomers and supercomplexes detected by BN-PAGE (labelled on the left hand side of the blot). We have added the BN-PAGE results obtained for complexes III and V in addition to complexes I and IV (new Fig S2 and updated figure 2) and modified the text in the manuscript accordingly.

• Figure 1 E would be more useful if the points were made 50-80% transparent so that all groups can be visualized. Otherwise here it just looks like a big black blob.

We thank the reviewer for this feedback. For clarity, we have replaced our Aspect ratio/ Form factor graph with bar graphs.

• Line 139: should read “interferon-stimulated”.

We have corrected this typo in the manuscript.

• Why did the authors use Mus terricolor instead of Mus spretus as the xenomitochonrial model? In cybrids, Mus spretus showed the most activation of IFN pathway. Is it because Mus terricolor is relatively divergent?

We have not succeeded in producing the mtDNA(Mus spretus) xenomouse line, only a single male germline animal (McKenzie et al 2004). We agree it would be interesting to investigate interferon response in this construct if it were available. However the Mus terricolor cybrid also displayed significant upregulation of the IFN pathway, and the availability of the corresponding mtDNA(Mus terricolor) xenomouse allowed investigating the IFN response in vivo in this construct. For clarity, we have included the following text in the manuscript: “(…). Although MusSpretus showed the highest activation of the interferon pathway, we did not succeed in producing the corresponding xeno mouse line (10).”

• Could you comment on the kinetics of IFNs upon infection, is there a peak window post-viral infection that may have been missed while assessing the IFN expression or peripheral response? Providing this information would enhance the impact of this paper and help guide subsequent research on this topic.

We thank the reviewer for allowing us to provide more details about type I IFN secretion for this set of experiments. Systemic HSV infection leads to high type I IFN production at 16 hrs with levels back to background at 2d post infection (Rasmussen et al., Journal of Virology 2007). However, local HSV infection does not lead to direct dissemination of virus to the draining LN (Whitney et al., J Virol 2018), therefore the kinetic for type I IFN secretion at this immune site will be delayed in comparison to a systemic infection. The peak of type I IFN secretion for this type of infection is not known. Therefore, we chose d2 post infection for analysis of type I IFN due to a previous study that demonstrated type I IFN secretion at this time point in a localized HSV infection model (Swiecki et al., PLOS pathogen 2013). We 

have amended Fig 3D, which now includes a naïve mouse as negative control. This clearly demonstrates that we have not missed type I IFN induction even though we might not have caught the peak.

Following HSV skin infection, the viral titer peaks on d2 post infection (primary site) and at days 4 and 5 (secondary site) in the skin (van Lindt et al., JI 2004). Innate immune responses such as type I IFNs will impact the early control, as shown by the lack of pattern recognition receptors on dendritic cells (Davey, JI 2010), whereas adaptive immunity will take several days to develop to impact viral proliferation (van Lindt, JI 2004). To get an indication as to whether innate or adaptive immunity is altered in xeno mice, we analysed viral titers at this early time point.

We have amended the text accordingly as follows:

Results section:

 (…) “As expected, ISG expression was increased following HSV-1 infection of control mice relative to naïve negative control (Fig 3D).”

Figure legend:

“ISG expression was measured in one naive control mouse (Control -HSV-1) to confirm interferon activation upon infection.”

• Could age be a factor in the observed IFN response along with genetic background? What was the cytokine or IFN profile-like in the older mice (line 163) compared to the few weeks- old mice used in the study (line 288)?

We agree it will be interesting to pursue this question, as we state in the manuscript (l. 228): “Additional work is required to understand how the interferon response may change in different tissues of the xeno mouse when aging or other relevant stressors are imposed.” However, we do not have available tissue from aged xenomice to explore this currently.

• Were there changes in mtDNA copy number in the model systems used relative to the wild-type strains, at baseline and post viral challenge?

Although we agree with the reviewer that this could be an interesting measurement, we did not measure mtDNA copy number in the xeno mouse at baseline or post viral challenge. We are no longer able to conduct this measurement in tissue samples used to generate the data in our manuscript, as we would not be able to determine mtDNA copy number at baseline post hoc. Whilst interesting, we don’t believe these data would affect the conclusions of our work and hope to address this question in future work. 

• Can the authors speculate on the potential translatability of these findings to patients with mitochondrial disease or mitochondrial replacement therapy? There are large inter-individual differences in immune responses in healthy individuals. To what extent do the author think mtDNA-nDNA interactions may account for inter-individual differences in healthy humans?

This is an interesting point raised by the reviewers, which we have discussed in the text as follows:

“The pathogenesis of mitochondrial diseases is traditionally ascribed to bioenergetic failure or oxidative stress. By revealing an unexpected link between mtDNA genotype and the interferon response, our work offers a new perspective on mitochondrial diseases where an increased susceptibility to viral infection is well known. Further work in models of both non-pathogenic and disease-causing mtDNA variants is warranted to investigate potential effects on the interferon immune response, as we speculate that nuclear-mitochondrial interactions may contribute to inter-individual differences seen in immune response between healthy humans.”

6. PLOS authors have the option to publish the peer review history of their article (what does this mean?). If published, this will include your full peer review and any attached files.

Do you want your identity to be public for this peer review? For information about this choice, including consent withdrawal, please see our Privacy Policy.

Reviewer #1: Yes: Kalpita Karan, Martin Picard

We agree to including the peer review history of our manuscript

We thank again the reviewers and the Editor for your consideration.

---

## [Decision Letter · Decision Letter 1]

15 Sep 2020

Nuclear response to divergent mitochondrial DNA genotypes modulates the interferon immune response

PONE-D-20-16529R1

Dear Dr. Sanchez,

We’re pleased to inform you that your manuscript has been judged scientifically suitable for publication and will be formally accepted for publication once it meets all outstanding technical requirements.

Kind regards,

Deepak Shukla

Academic Editor

PLOS ONE

Additional Editor Comments (optional):

Reviewers' comments:

Reviewer's Responses to Questions

**Comments to the Author**

1. If the authors have adequately addressed your comments raised in a previous round of review and you feel that this manuscript is now acceptable for publication, you may indicate that here to bypass the “Comments to the Author” section, enter your conflict of interest statement in the “Confidential to Editor” section, and submit your "Accept" recommendation.

Reviewer #1: All comments have been addressed

2. Is the manuscript technically sound, and do the data support the conclusions?

Reviewer #1: Yes

3. Has the statistical analysis been performed appropriately and rigorously? 

Reviewer #1: Yes

4. Have the authors made all data underlying the findings in their manuscript fully available?

Reviewer #1: Yes

5. Is the manuscript presented in an intelligible fashion and written in standard English?

Reviewer #1: Yes

6. Review Comments to the Author

Reviewer #1: All comments have been addressed and the paper is significantly improved. This paper fits nicely into a burgeoning field of mito-nuclear interactions and modulation of innate and adaptive immune processes - this paper is a valuable addition to this field.

7. PLOS authors have the option to publish the peer review history of their article (what does this mean?). If published, this will include your full peer review and any attached files.

Reviewer #1: **Yes: **Martin Picard

---

## [Editor Report · Acceptance letter]

28 Sep 2020

PONE-D-20-16529R1 

Nuclear response to divergent mitochondrial DNA genotypes modulates the interferon immune response 

Dear Dr. Lopez Sanchez:

I'm pleased to inform you that your manuscript has been deemed suitable for publication in PLOS ONE. Congratulations! Your manuscript is now with our production department. 

Kind regards, 

on behalf of

Prof. Deepak Shukla 

Academic Editor

PLOS ONE